# Dynamic Characteristics of Plasma in Ultrasonic-Assisted Narrow-Gap Laser Welding with Filler Wire

**DOI:** 10.3390/ma16020502

**Published:** 2023-01-04

**Authors:** Ren Wang, Zhenxing He, Xiaoyang Kan, Ke Li, Fugang Chen, Juan Fu, Yong Zhao

**Affiliations:** 1CRRC Qingdao Sifang Co., Ltd., Qingdao 266111, China; 2Provincial Key Lab of Advanced Welding Technology, Jiangsu University of Science and Technology, Zhenjiang 212003, China

**Keywords:** narrow-gap laser welding with filler wire, ultrasonic assisted, plasma morphology, plasma electron density, plasma temperature

## Abstract

Laser welding with filler wire was applied to Q345D in a narrow gap under ultrasonic assistance, and the dynamic characteristics of plasma were studied by high-speed imaging and spectral acquisition. The results showed that the plasma area decreased gradually with increasing distance between the ultrasonic loading position and welding seam. The electron density and temperature of the plasma with ultrasonic assistance were higher than those without ultrasound. The electron density was approximately 10^16^~10^17^ cm^−3^, and the plasma temperature was approximately 4000~6000 K. Ultrasonic assisted laser wire filling welding can bring about cavitation effect and significantly reduce the porosity problem.

## 1. Introduction

Improving the efficiency and quality of thick-plate welding has always been an emphasis of welding experiments [1,2,3]. Large grooves are required for thick plates in traditional welding methods, which leads to an increase in the filling quantity of welding wire and consequently to an increase in cost. Laser welding with filler wire is a method in which a laser served as the heat source. The base metal was melted by one part of the laser heat and formed the molten pool [4,5,6,7,8,9]. The welding wire was melted by another part, formed the droplet, and consequently transited into the molten pool. Compared with traditional welding methods, narrow-gap welding sharply decreases the filling quantity and enhances the efficiency. For high-strength steel, narrow-gap laser welding can meet the requirements of processing and assembly precision of the component welding application and, at the same time through the filling of welding wire, can carry out metallurgical control on the weld, refine the microstructure grain, and can use a low-power laser to achieve the welding of a medium and thick plate so as to reduce the dependence on high-power lasers [4,10,11,12,13,14,15].

At present, the addition of ultrasonic-assisted welding has become a vital regulation method. Due to the high quality of welding and fewer pores, an increasing number of researchers have introduced ultrasound into laser welding [16,17,18,19,20,21,22].

Zhang et al. [23] analyzed the metal plastic deformation state at the welding interface by using the finite element method. The results showed that the welding form of the metal interface was that the metal was subjected to shear stress by ultrasonic vibration. The friction heat generated would reduce the mechanical properties of the metal, which increased the plastic-deformation degree of the local metal and released more heat. C.Y. Kong et al. [24] introduced an ultrasonic roll welding and bonding device in the multilayer connection of 6061 aluminum foil. Under the condition of power P = 3.3 kW and frequency F = 20 kHz, the amplitude, contact pressure, and welding speed were adjusted to realize the reliable connection of 6061 aluminum foil after 100 microns. Sibillano et al. [25] studied the plasma dynamic and static behavior in laser welding by spectrograph and found that with increasing laser power, the weld depth increased, and the plasma temperature decreased.

In this study, Q345D was welded by ultrasonic-assisted narrow-gap (3 mm gap width) laser welding with filler wire, and the plasma was observed via a high-speed camera and spectrograph. The dynamic characteristics of plasma were analyzed under different ultrasonic loading conditions.

## 2. Material and Experimental Procedures

The basic material used in this work was a Q345D steel plate with dimensions of 200 mm × 100 mm × 12 mm. The corresponding filler wire was ER50-6 with a diameter of 1.2 mm. Table 1 shows the chemical composition of the base material and the filler wire. In this research, the welding system was composed of an IPG fiber laser machine (YLS-6000, American IPG Photonics company, Oxford, MA, USA), an ABB robot (Asea Brown Boveri in Zurich, Switzerland), and a Fronius TPS4000 welding machine (Fronius, Pettenbach, Austria). The laser power was 5 kW, and the defocused length (Δf) was 30 mm. The wire feeding speed (Vf) was 4 m/min, the welding speed (Vw) was 0.58 m/min, and the height of the welding wire (HLW) was 1 mm. Argon shielding gas (99.99%) was applied at a flow rate of 15 L/min to protect the weld. This study investigated the influence of ultrasonic loading position on plasma morphology.

During the welding process, a high-speed camera (CP80-3-M-540, made by Optronis, Kehl, Germany) and a spectrograph (MX2500+, made by Ocean Optics, Orlando, FL, USA) were used in this research. Plasma was difficult to measure directly, so the temperature and electron density could only be obtained by the collected spectral information. The schematic of the experimental setup is shown Figure 1.

After welding, MATLAB was used to process and analyze the images that were shot by a high-speed camera, and then, the characteristics of the plasma area under different conditions were extracted. Moreover, the obtained spectral information was used to compute the temperature and electron density of the plasma.

## 3. Acquisition of Experimental Information

### 3.1. Acquisition of Plasma Area

With the help of a high-speed camera, the plasma shape can be clearly observed, as shown in Figure 2. It can be divided into three parts: (1) background, (2) outer peripheral region of plasma, and (3) core region of plasma. The third region was the research object of this study.

In the gray image, the change of light and shade is expressed according to a total of 256 gray scales from 0 to 255, in which 0 is completely black, 255 is completely white, and other values are gray in between. Each high-speed photography image contains 512 × 512 pixels, with a total of 262,144 pixels. The software is used to calculate the frequency of each gray value, as shown in Figure 2b, and most gray values are below 100, which is considered as the background, 100–250 as the shell plasma and metal vapor, and 250–255 as the plasma core region. The actual size of each 512 × 512 px high-speed photographic image is 18.6 × 18.6 mm. Therefore, the actual area of the plasma core region can be calculated only by using software to count the number of pixels in the core region.

In order to reflect the synergistic mechanism of ultrasonic and laser, it is necessary to calculate the fluctuation of plasma area during welding. Based on the above calculation ideas, the matlab algorithm was designed to calculate no less than 1500 images, as shown in Figure 3.

### 3.2. Acquisition of Plasma Spectral Information

In this study, a spectrograph was used to gather the groove of narrow-gap laser wire filling welding. As shown in Figure 4. The whole spectrum information-acquisition system consists of an eight-channel spectrometer and a set of optical amplification systems. The incident light is transmitted to the optical fiber probe by the amplification system. The spectrometer splits the light into eight channels according to the wavelength and records the spectral information of all channels. Optical fiber probes are set in the center of the image screen to capture spectral signals at different positions in the plasma.

The groove width used in this experiment is 3 mm, and the welding wire with a diameter of 1.2 mm is in the middle of the groove.

## 4. Results and Discussion

### 4.1. Plasma Morphology Analysis

The plasma high-speed photographic images under different conditions are shown in Figure 5. With the increase in DUL (distance from ultrasonic loading position to weld), the height and area of plasma decrease gradually. This is because the vibration and stirring effect of ultrasonication on the molten pool became weaker, and the excitation effect on the plasma also weakened gradually. When DUL = 20 mm, the variation period of plasma was approximately 5.25 ms, which increased by approximately 40% compared with DUL = 10 mm. However, when DUL = 30 mm, due to the weakening of the ultrasonic effect, the variation period was not obvious, and it was more similar to the case without ultrasonication. The plasma was more concentrated in the groove, which was difficult to erupt.

The maximum area and area variation frequency of plasma under different parameters are shown in Figure 6. The farther the ultrasonic position was from the weld, the smaller the plasma area was. On the one hand, the propagation process of ultrasound results in energy attenuation due to the extension of the path; on the other hand, the grain boundary in the material reflects the ultrasound, which is opposite to the propagation direction of ultrasound, increasing the loss of ultrasonic energy. When DUL = 10 mm, the plasma had a maximum area of 125 mm^2^. When DUL = 20 mm, the area was 100 mm^2^. However, the plasma occupied only half the area in the absence of ultrasound.

The change in plasma shape in the narrow-gap groove by ultrasonic impact is mainly caused by the cavitation effect and acoustic flow effect on the molten pool. As the DUL increased, the plasma morphology tended to be that without ultrasonic assistance, and its area gradually decreased. This indicates that the effect of ultrasonic shock on the molten pool is also gradually reduced, the energy released by the rupture of the keyhole in the molten pool was correspondingly lower, and the plasma volume of keyhole eruption was increasingly lower. This phenomenon was due to the attenuation of energy caused by the extension of distance in the propagation process of ultrasound. This was also because the grain boundaries in the material reflected the ultrasonic wave, which was opposite to the ultrasonic propagation direction and increased the ultrasonic energy loss. Meanwhile, if the ultrasonic loading distance was too close, the plasma would change sharply, which led to poor stability.

### 4.2. Plasma Electron Density

According to the spectral information collected from the paths and collection points set before, the electron density under different DUL conditions is shown in Figure 7 after calculation. When DUL = 30 mm, the effect of ultrasound was weak. As the distance from the bottom of the groove increased, the electron density decreased more rapidly and obviously. The main reason was that the vibration effect of laser energy on the molten pool was weakened. When the distance decreased, the cavitation effect was weakened, the energy release of the molten pool was reduced, and the eruption of keyhole plasma was much less. When DUL = 10 mm and DUL = 20 mm, the difference between them was not obvious. The reason was that the collection distance was not long enough, and the effect between them was not obvious. In addition, the vibration of the molten pool was relatively small due to the lack of ultrasound loading, which led to difficulty in plasma spreading and tended to concentrate near the top of the molten pool. Overall, the plasma electron density in the groove of narrow-gap laser welding with filler wire was approximately 1016 cm^−3^ orders of magnitude. The addition of ultrasound did not cause a qualitative change in the electron density, but it expanded the different positions of the electron density variation trend, which made the distribution gradient more apparent. The effect became more obvious when the ultrasound became closer to the weld.

### 4.3. Plasma Temperature

According to the spectral information collected from the paths and collection points set before, the plasma temperature under different DUL conditions is shown in Figure 8 after calculation. The general variation trend of plasma temperature was similar to that of electron density. With the increase in the height from the bottom of the groove, the temperature gradually decreased, and the approximate range was between 4000 and 6000 K. The closer the ultrasonic position was to the weld, the higher the plasma temperature was. The main reason was that the closer the ultrasonic source was to the weld, the stronger the cavitation effect. At this point, the bubbles and tiny pores in the weld pool rapidly expanded and then closed instantly, releasing a large amount of energy and forming local high pressure in the keyhole to accelerate the outwards eruption of plasma and increase the temperature.

When it was without ultrasound and DUL = 30 mm, different high temperature variation trends were slowing, and a slight temperature fluctuation emerged, as shown in Figure 8b. As the plasma erupted more violently when DUL = 10 mm, and more plasmas were located in the external region of the groove, the distance was further away from the bottom of the groove when DUL = 20 mm. It influenced the plasma temperature variation trend at different locations by influencing the plasma eruption degree.

The four spectral lines in Table 2 were used for Boltzmann fitting to calculate the plasma temperature. The fitting results are shown in Figure 9.

### 4.4. Influence of Ultrasonic Assistance on Weld Forming

Based on previous process research, a 12 mm thick Q345D steel plate was welded by laser wire filling welding under ultrasonic-assisted and non-ultrasonic conditions. The welding parameters are shown in Table 3. The distance between the ultrasound and welding center was 20 mm. The cross-sectional macrograph of the joint is shown in Figure 10 and Figure 11. According to the “GB/T 3323-2005 Radiographic examination of fusion welded joints in metallic materials”, X-ray testing was conducted on the weld area, and the results are shown in Figure 12.

The porosity generated in the process of thick-plate narrow-gap laser wire filling welding is mainly due to the instability of deep penetration keyhole, which leads to uneven concatenation at the junction of welding area and base material and also causes the protective gas to be involved with the beam, forming bubbles in the sag, and the gas cannot escape regularly, thus forming technological porosity. However, there may also be a small number of hydrogen pores, solidified porosity pores, and residual pores due to the inability of metal vapor to escape at low temperature. During welding, ultrasonic cavitation makes the tiny bubbles in the molten pool vibrate, which will help the bubbles escape from the molten pool.

By comparing Figure 10 and Figure 11, it can be seen that ultrasound-assisted laser wire filling welding can obtain a deeper penetration morphology. Due to the addition of ultrasound, the molten pool is affected by cavitation, and the plasma area increases, indicating that the addition of ultrasound makes it easier for laser energy to propagate downwards. At this time, the electron density and temperature were higher, and the laser energy was more easily spread down, resulting in an increase in the penetration. Figure 12 shows that the porosity of ultrasound-assisted welding was reduced by 50% compared with that of welding without ultrasound. In the welding process, ultrasonic high-frequency vibration agitated the molten pool more violently, and the bubbles vibrated and closed promptly after rapid expansion. Therefore, the number of pores in the weld with ultrasound assistance was much less than that without ultrasound assistance.

### 4.5. Influence of Ultrasonic aid on the Microstructure and Hardness of Weld

The microstructure morphology analysis and hardness test were carried out for metallography with or without ultrasonic impact. The microstructure is shown in Figure 13. The hardness distribution is shown in Figure 14. In the hardness test, the interval between the points is 0.3 mm, where BM represents the base metal, HAZ represents the heat-affected zone and WB represents the weld bead.

It can be seen from the figure that ultrasonic assistance can significantly refine the grain and improve the weld hardness, which is mainly due to the cavitation effect and the sound flow effect caused by ultrasonication in the molten pool, which changes the growth mode and direction of the grain and refines the grain. The ultrasonic cavitation will produce cavitation bubbles in the molten pool. After the rapid expansion and rapid rupture of the cavitation bubble, it releases a large amount of energy in a very short moment, resulting in a strong impact in the small local area of the molten pool, breaking the surrounding grains, increasing the nucleation point, and improving the nucleation rate. The sound flow effect speeds up the convection of liquid metal, which makes the fine crystal nuclei broken by the cavitation effect more evenly distributed in the molten pool and accelerates the process of nonuniform nucleation, so the microstructure of the weld zone is refined.

## 5. Conclusions

From the above investigations, the main conclusions can be summarized:(1)The plasma area, electron density, and temperature with ultrasound-assisted narrow-gap laser wire filling welding were higher than those without ultrasound, but as the distance between the ultrasound and weld increased, the plasma area, electron density, and temperature showed a declining trend;(2)When DUL = 10 mm, the area distribution was very discrete. When DUL = 20 mm, the plasma area distribution was more concentrated, and the morphology was more stable compared with DUL = 10 mm. When DUL = 30 mm, the plasma area was smaller than that of DUL = 20 mm, but the dispersion degree was higher. When ultrasound was not added, the plasma morphology was significantly smaller, the dispersion degree was lower, and the area distribution was more concentrated than that under other conditions;(3)The plasma electron density of laser wire filling welding in the narrow-gap groove was approximately 1016 cm^−3^. The addition of ultrasound did not cause a qualitative change in electron density, but it increased the variation trend of electron density at different positions. Moreover, the closer the ultrasound was to the weld, the more obvious the effect was;(4)With increasing distance from the bottom of the groove, the plasma temperature decreased gradually, and the approximate range was between 4000 and 6000 K;(5)Ultrasonic-assisted laser wire filling welding can produce cavitation and acoustic flow effects, significantly reduce the porosity problem, refine the grain, and improve the mechanical properties of welded joints;(6)In this study, ultrasonic-assisted laser wire filling welding technology was used to successfully weld a 12 mm Q345D thick plate with a narrow-gap groove. The welding quality was good, and the porosity was low, which improved the theoretical basis for the welding manufacturing of medium thick plates in the field of shipbuilding.

## Figures and Tables

**Figure 1 materials-16-00502-f001:**
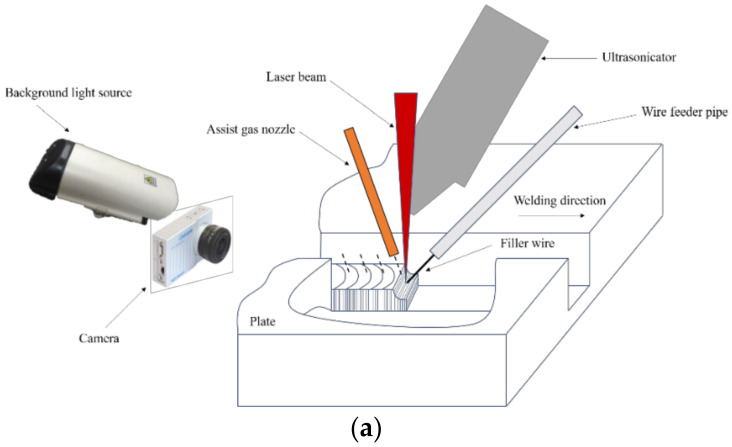
Schematic of the experimental setup: (**a**) High-speed photographic equipment; (**b**) laser plasma spectral analysis system.

**Figure 2 materials-16-00502-f002:**
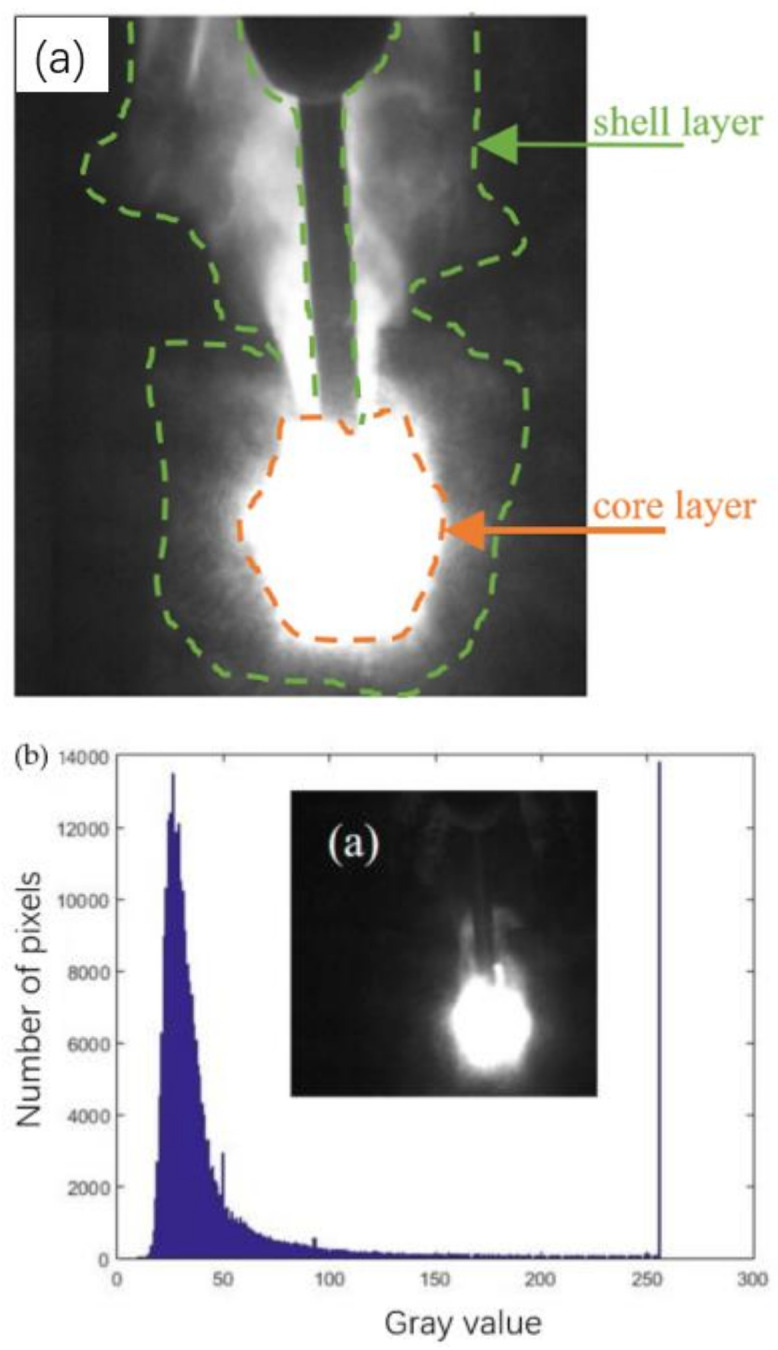
Plasma morphology at high speed. (**a**) Schematic diagram of different layers in plasma images; (**b**) High-speed image of plasma and (a) gray distribution.

**Figure 3 materials-16-00502-f003:**
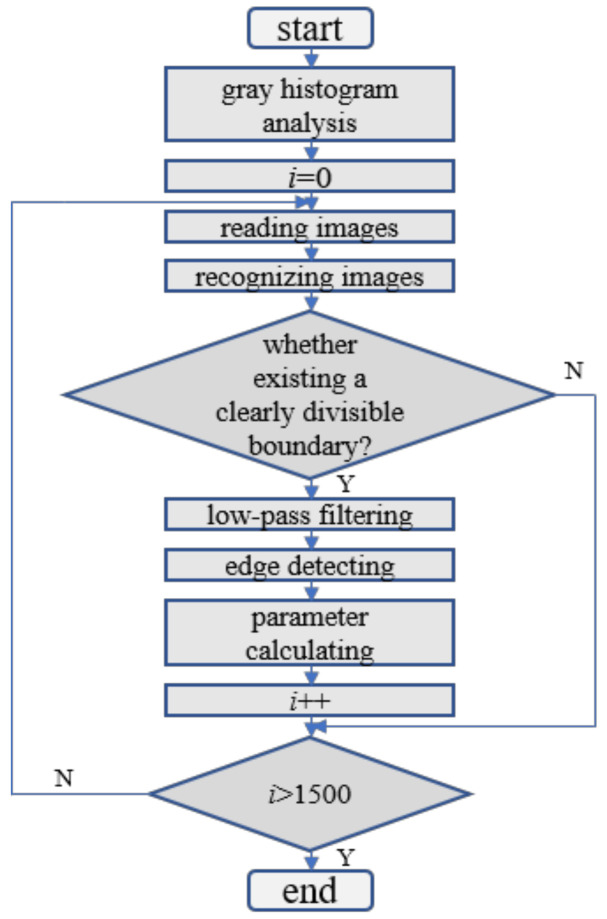
Area calculation of the plasma from high-speed photography.

**Figure 4 materials-16-00502-f004:**
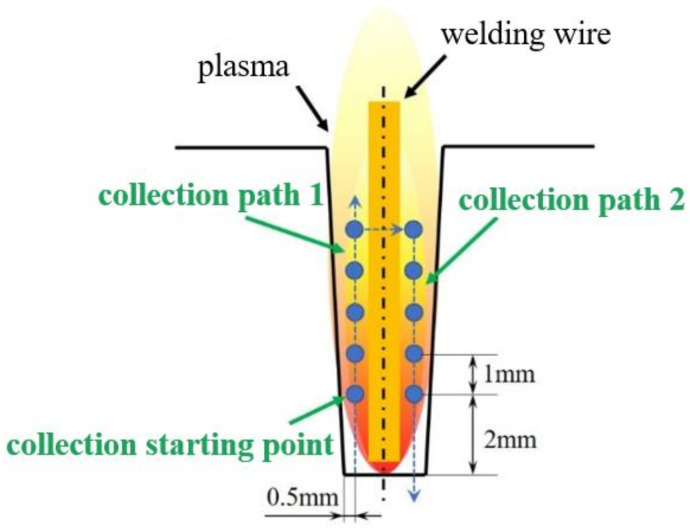
Schematic diagram of spectral acquisition.

**Figure 5 materials-16-00502-f005:**
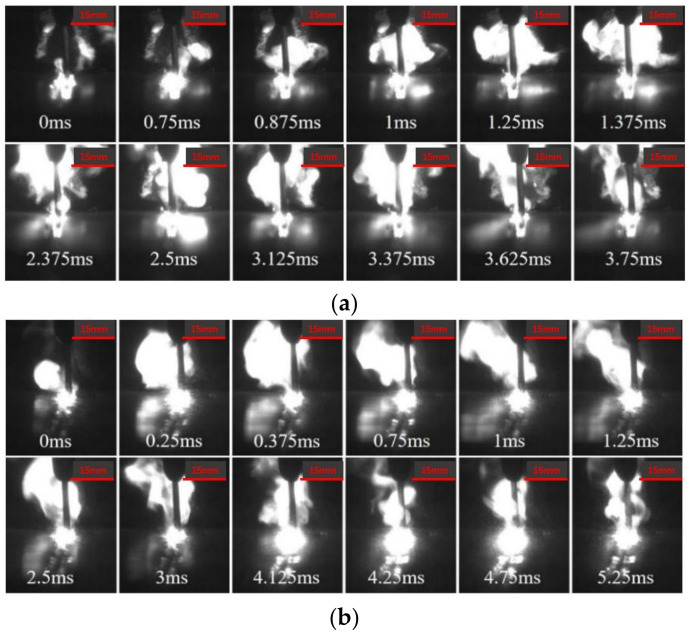
Plasma image under different DUL. (**a**) DUL = 10 mm; (**b**) DUL = 20 mm; (**c**) DUL = 30 mm; (**d**) without ultrasonication.

**Figure 6 materials-16-00502-f006:**
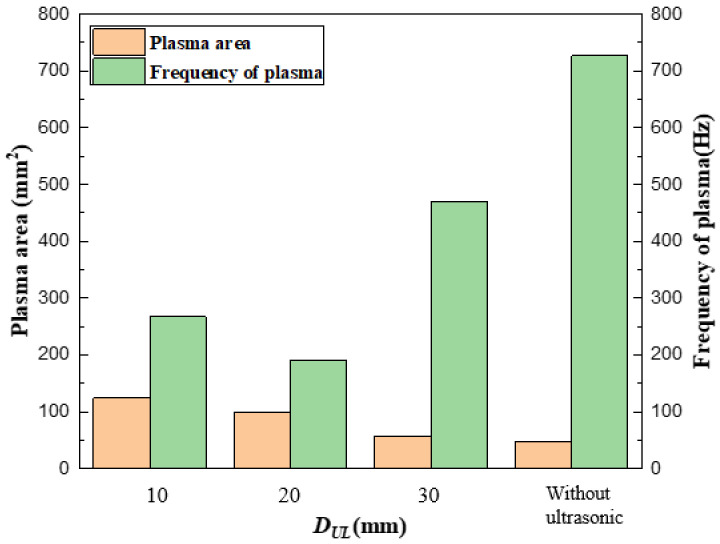
The relationship between the area and frequency of plasma and DUL.

**Figure 7 materials-16-00502-f007:**
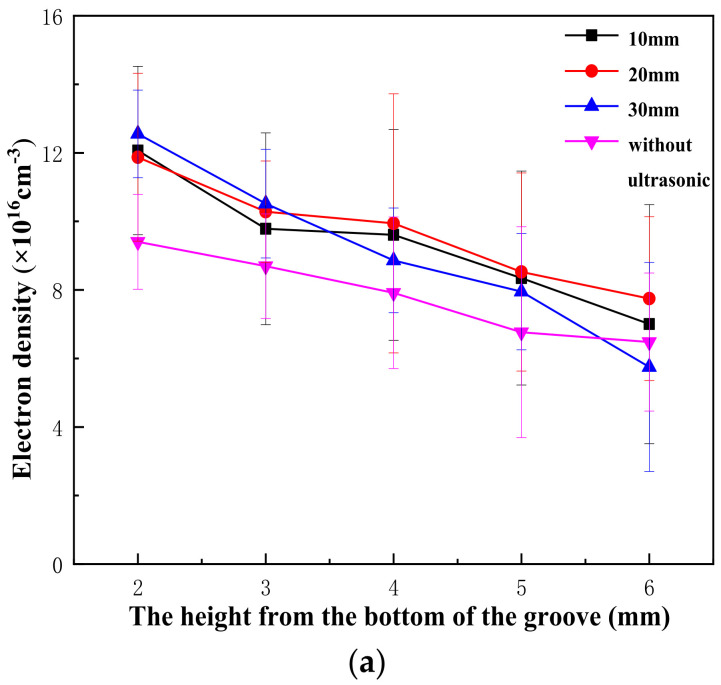
Distribution of electron density in plasma under different DUL: (**a**) path 1; (**b**) path 2.

**Figure 8 materials-16-00502-f008:**
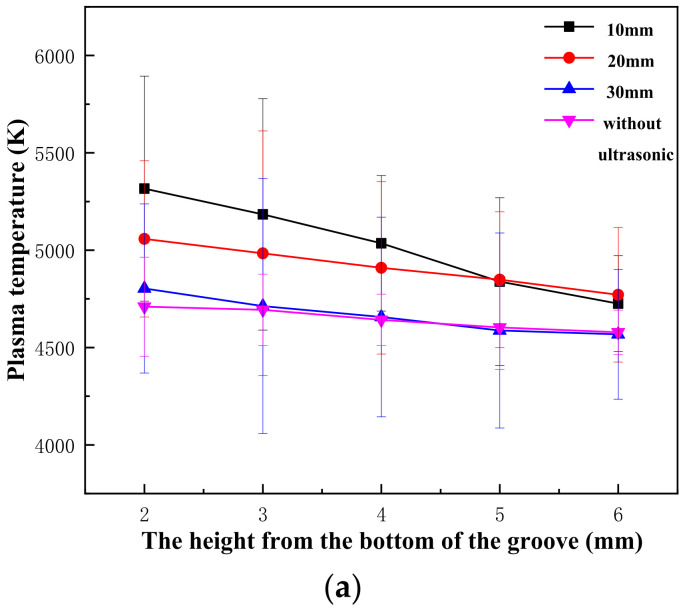
Distribution of plasma temperature under different DUL: (**a**) path 1; (**b**) path 2.

**Figure 9 materials-16-00502-f009:**
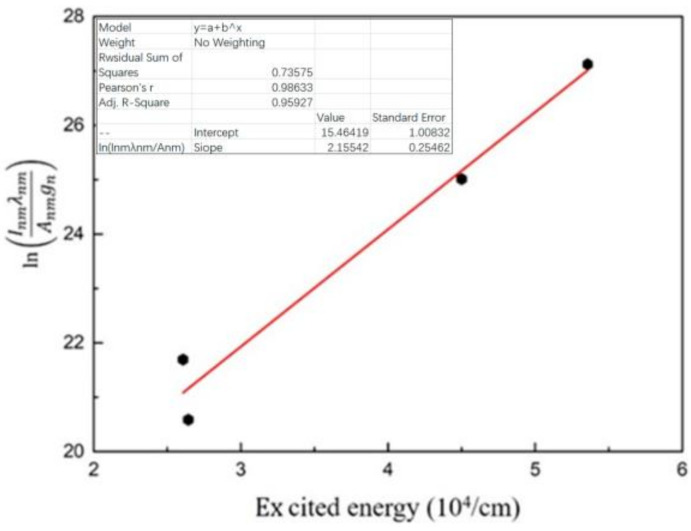
Boltzmann maps and their fitting line for calculating electron temperature.

**Figure 10 materials-16-00502-f010:**
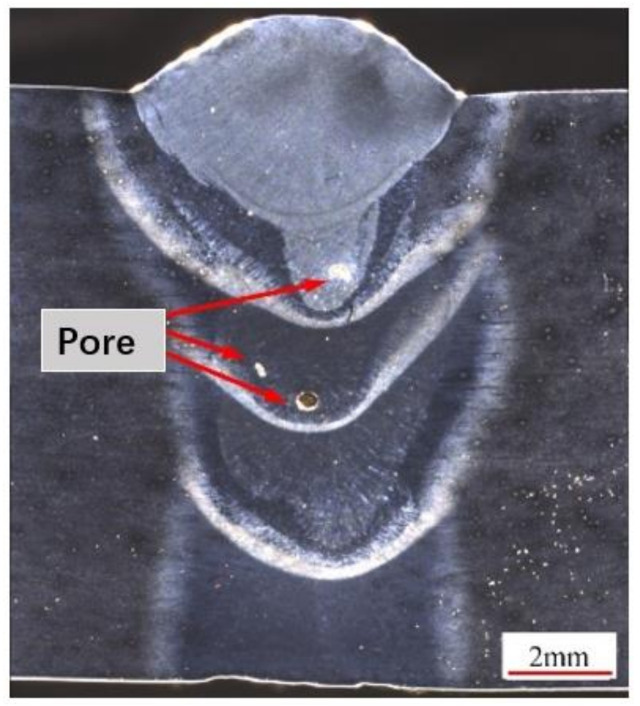
Cross-section of the narrow-gap joint without ultrasonication.

**Figure 11 materials-16-00502-f011:**
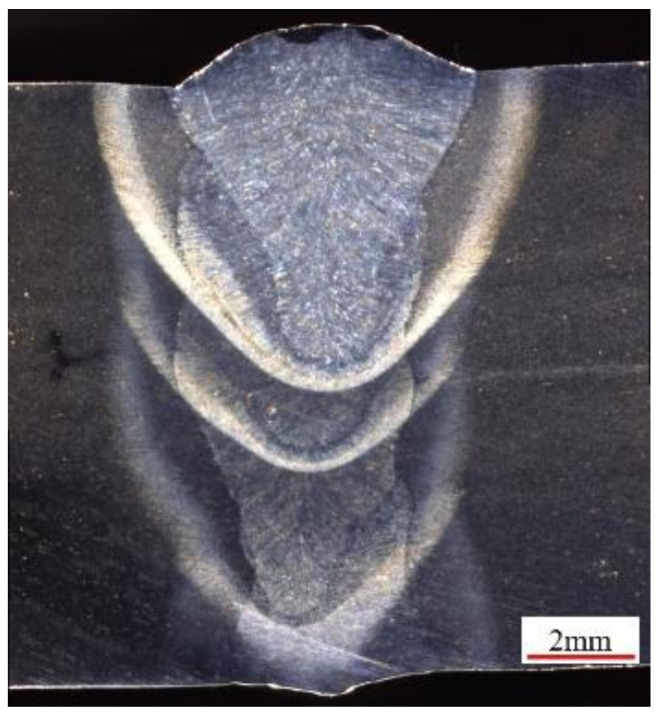
Cross section of narrow-gap joint with ultrasonic.

**Figure 12 materials-16-00502-f012:**
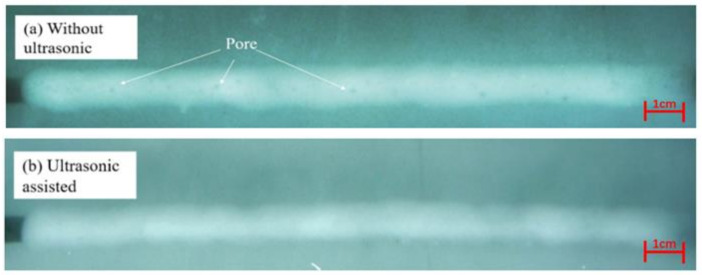
X-ray inspection results of 12 mm thick Q345D welded by narrow-gap laser welding with filler wire: (**a**) without ultrasonication; (**b**) ultrasound-assisted welding.

**Figure 13 materials-16-00502-f013:**
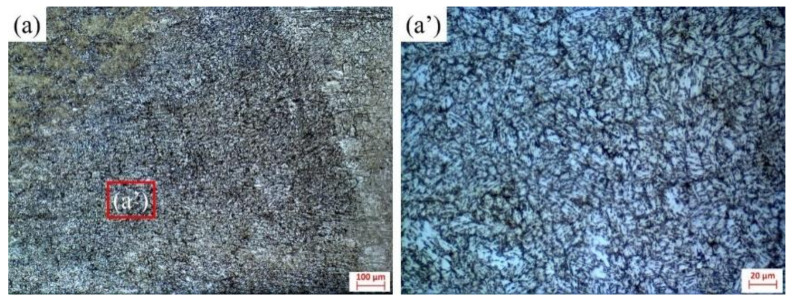
Microstructure of weld bead center: (**a**,**a’**) without ultrasonic assistance; (**b**,**b’**) with ultrasonic assistance.

**Figure 14 materials-16-00502-f014:**
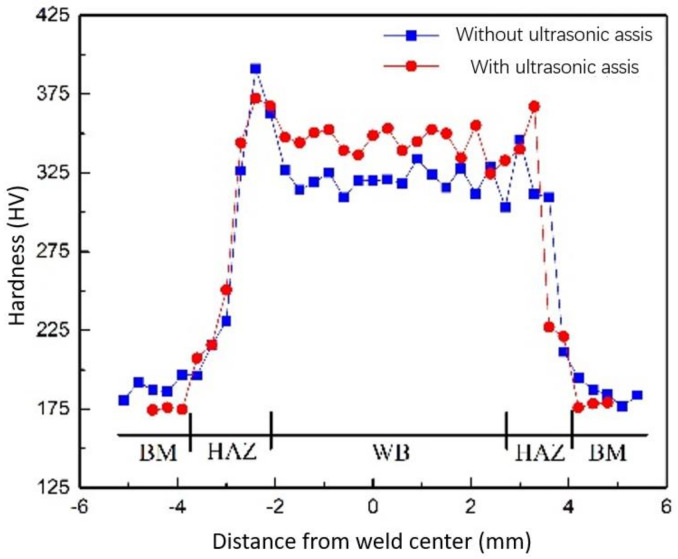
Hardness distribution of the welded joints.

**Table 1 materials-16-00502-t001:** Chemical composition of the Q345D base metal and ER50-6 filler wire (wt%).

Material	C	Mn	Si	P	S	Cu	V	Al
Q345D	≤0.20	≤1.7	≤0.55	≤0.025	≤0.025	≤0.3	0.02–0.15	≥0.015
ER50-6	0.06–0.15	1.4–1.85	0.80–1.15	≤0.025	≤0.035	≤0.5		

**Table 2 materials-16-00502-t002:** Fe atomic emission lines selected in the calculation of electron temperature (Data from the NIST Atomic Spectra Database Lines Form).

Wave Length (nm)	*E_n_* (cm^−1^)	*E_n_* (eV)	*g_n_*	*A_nm_* (10^8^ s^−1^)
537.149	26,339.696	3.2675	5	0.0105
538.337	53,352.989	6.6149	13	0.781
539.317	44,677.000	5.5392	9	0.0491
539.713	25,899.989	3.2112	9	0.00258

**Table 3 materials-16-00502-t003:** Welding process parameters.

Layer Number	Power (kW)	Welding Speed (m/min)	Wire Feeding Speed (m/min)	Defocus Length (mm)
1	5.8	0.36	1.5	+10
2–3	5.0	0.48	4.0	+20
4	5.8	0.36	3.5	+40

## Data Availability

The data used to support the findings of this study are available from the corresponding author upon request.

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
