# Peer review of "Dynamic Characteristics of Plasma in Ultrasonic-Assisted Narrow-Gap Laser Welding with Filler Wire"

_materials, 2023, doi:10.3390/ma16020502_

Round 1

Reviewer 1 Report

Good paper. Major revisions are in order for the authors to address the comments detailed in the points below.

Language needs to be polished. Several minor mistakes found.

“Improving the efficiency and quality of thick plate welding had always been an emphasis of welding experiment”: references needed. See for example 10.1016/j.matdes.2022.111176 and 10.1016/j.scriptamat.2022.115053 and complement.

“Thus, laser wire filling welding in narrow gap was a very efficient method”: for all metals? Clarify.

“sisted narrow-gap laser weld”: how to define narrow?

“asma clutter was quite serious by the high-speed ca”: what does this mean? Unclear.

“MATLAB was executed automatically according to the pre-prog”: more details on this are needed to be provided.

More details on the plasma spectral acquisition are also needed.

Fig 5 needs a scale.

“be seen that the farther the ultrasonic position was from the weld, the smaller the plasma area wa”: and why is that? Needs to be discussed.

“n was that the vibration effect of laser energy on the molten pool was weakened”: how to confirm this?

How are the errors bars of fig 7 obtained? Detail please.

Where were the Fe atomic emission lines obtained from? Must be detailed.

Fig 10: what is the pore formation mechanism? In fig 11 looks like there are pores on the top of the welded joint. Clarify please.

Fig 12 needs a scale.

“The eruption of plasma and metal vapor was more intense, and the vapor jet force was larger”: and why is that?

“In the welding process, the molten pool was stirred”: but in fusion welding there is always stirring. See 10.1016/j.matdes.2022.110717 and complement.

Have the authors checked for the changes in microstructure and hardness using the two variants? This should be comments on because there are certainly changes.

“Due to the ultrasonic cavitation, the weld surface was more uniform and smoother, the penetration of ultrasonic assisted laser wire filling welding was larger and the porosity was reduced by 50%”: can one achieve fully defect-free parts?

Reviewer 2 Report

Due to the high quality of welding and the less pore, the authors of this manuscript introduced ultrasound into laser welding.  In presented study, Q345D was welded by ultrasonic-assisted narrow-gap laser welding with filler wire, and the plasma was observed via high-speed camera and spectrograph. The dynamic characteristics of plasma was analyzed under different ultrasonic loading  conditions. The presented solution to the problem of obtaining high-quality connections is very important for the Technology of Ship Project and it can be used there.

1.       The work shows an adequate understanding of the literature on the subject, but the authors cite literature sources without the names of the authors. I propose to improve the literature list.

2.       Not all research results are clearly presented. I am asking the authors to explain:

·       How the X-ray inspection was carried out, please describe the test method or state the standard.

·       I suggest adding one or two sentences in the analysys too, because the  authors wrote, that “due to the addition of ultrasonic, the weld surface was more uniform and smoother, the penetration was larger and the porosity was reduced by 50% “ (line 14-15) but figures 11-12 (line 224-226) only showed the results without porosity (maybe the porosity was reduced by 100% in this case)

·       please explain the differences between the notations DUL and DUL (text and figures) and please describe DUL abbreviation (line 108-109)

3.       The analyse can be improved. I suggest a minor revision:

·       The results, presented at the figure 9 should be corrected. Please, describe the axes

·       Please correct the entries using indices, e.g. 7cm-3, En (cm-1)

·       Please enlarge picture 10, or insert a picture of better quality (scale)

4.       I suggest adding one or two sentences in the conclusion. Please write an application example of the presented study and a suggested direction of future research.

The article bridges the gap between theory and practice. Research is still underway to improve the process of plasma under different ultrasonic loading  conditions. It is important to optimize the process parameters in order to obtain the correct properties of the joint. The presented research results are impressive.

Round 2

Reviewer 1 Report

Several comments not properly addressed. 

Comment 1: not addressed. See suggested references and modify accordingly.

Comment 2: too vague. Needs to be further detailed.

Comment 5: why the selection of those parameter ranges. Needs to be clarified.

Comment 8: needs to be further discussed.

Comment 12: needs to be further clarified. The answer is too vague.

Revise properly.
